# A cross-sectional analysis of meteorological factors and SARS-CoV-2 transmission in 409 cities across 26 countries

Francesco Sera[1,2 ✉], Ben Armstrong [1], Sam Abbott[3,4], Sophie Meakin[3,4], Kathleen O'Reilly [3,4], Rosa von Borries [5], Rochelle Schneider [1,6,7,8], Dominic Royé [9], Masahiro Hashizume [10,11,12], Mathilde Pascal[13], Aurelio Tobias[11,14], Ana Maria Vicedo-Cabrera [15,16], MCC Collaborative Research Network*, CMMID COVID-19 Working Group*, Antonio Gasparrini [1,6,17] & Rachel Lowe [3,4,6,18 ✉]

There is conflicting evidence on the influence of weather on COVID-19 transmission. Our aim is to estimate weather-dependent signatures in the early phase of the pandemic, while controlling for socio-economic factors and non-pharmaceutical interventions. We identify a modest non-linear association between mean temperature and the effective reproduction number ($R_e$) in 409 cities in 26 countries, with a decrease of 0.087 (95% CI: 0.025; 0.148) for a 10 °C increase. Early interventions have a greater effect on $R_e$ with a decrease of 0.285 (95% CI 0.223; 0.347) for a 5th - 95th percentile increase in the government response index. The variation in the effective reproduction number explained by government interventions is 6 times greater than for mean temperature. We find little evidence of meteorological conditions having influenced the early stages of local epidemics and conclude that population behaviour and government interventions are more important drivers of transmission.

[1] Department of Public Health, Environments and Society, London School of Hygiene & Tropical Medicine, London, UK. [2] Department of Statistics, Computer Science and Applications "G. Parenti", University of Florence, Florence, Italy. [3] Centre for Mathematical Modelling of Infectious Diseases, London School of Hygiene & Tropical Medicine, London, UK. [4] Department of Infectious Disease Epidemiology, London School of Hygiene & Tropical Medicine, London, UK. [5] Charité Universitätsmedizin, Berlin, Germany. [6] Centre on Climate Change and Planetary Health, London School of Hygiene & Tropical Medicine, London, UK. [7] Forecast Department, European Centre for Medium-Range Weather Forecast (ECMWF), Reading, UK. [8] Φ-Lab, European Space Agency, Frascati, Italy. [9] Department of Geography, CIBER of Epidemiology and Public Health (CIBERESP), University of Santiago de Compostela, Santiago de Compostela, Spain. [10] Department of Paediatric Infectious Disease, Institute of Tropical Medicine, Nagasaki University, Nagasaki, Japan. [11] School of Tropical Medicine and Global Health, Nagasaki University, Nagasaki, Japan. [12] Department of Global Health Policy, Graduate School of Medicine, The University of Tokyo, Tokyo, Japan. [13] Santé Publique France, Department of Environmental and Occupational Health, French National Public Health Agency, Saint Maurice, France. [14] Institute of Environmental Assessment and Water Research (IDAEA), Spanish Council for Scientific Research (CSIS), Barcelona, Spain. [15] Institute of Social and Preventive Medicine, University of Bern, Bern, Switzerland. [16] Oeschger Center for Climate Change Research, University of Bern, Bern, Switzerland. [17] Centre for Statistical Modelling, London School of Hygiene & Tropical Medicine, London, UK. [18] Barcelona Supercomputing Center, Barcelona, Spain. *Lists of authors and their affiliations appear at the end of the paper. ✉email: francesco.sera@lshtm.ac.uk; rachel.lowe@lshtm.ac.uk

Severe acute respiratory syndrome coronavirus 2 (SARS-CoV-2) has rapidly spread across the globe, traversing diverse climatic and environmental conditions. Sustained local transmission has occurred in most countries, leading to political, social and economic challenges and devastating loss of life. From the early phase of the pandemic, there has been speculation that weather conditions could modulate SARS-CoV-2 transmission patterns. The debate has been driven by analogy with existing seasonal endemic respiratory viral infections, such as influenza and other human coronaviruses, which tend to peak in the drier and colder winter months in temperate climates[1]. However, specific mechanisms behind this seasonality, in terms of host immunity and susceptibility, viral stability or weather-sensitive human behaviour are poorly understood[2]. Dynamic transmission modelling has shown that meteorological variables, such as temperature and humidity, are unlikely to have been a dominant transmission risk factor in the early stages of the COVID-19 pandemic, given high population susceptibility[3,4]. As SARS-CoV-2 is a new virus to humans, with <1 year of data available at the time of writing, ascertaining the potential for weather modulated transmission is challenging. Several studies have attempted such analyses. However, many such studies had methodological weaknesses and the results were at times conflicting[5,6]. Study findings for temperature resulted in either a positive[7,8], negative[9,10], non-linear[11,12] or non-significant association[13,14] with the COVID-19 response variable. For example, most studies did not control for key modulating factors, such as varying government restrictions, socio-economic indicators, population density or age structure[15–17].

In this study, we overcome methodological issues of previous approaches by using a two-stage ecological modelling approach to examine the impact of meteorological variables on SARS-CoV-2 transmission by comparing cities located across the globe, while accounting for confounding of non-pharmaceutical interventions (NPIs) and city-level covariates. The study is based on an extensive dataset, collected by the Multi-Country Multi-City MCC Collaborative Research Network (https://mccstudy.lshtm.ac.uk/), consisting of time series of daily COVID-19 cases registered between 11 January and 28 April 2020 in 409 locations (cities or small regions) in 26 countries. In the first stage, we estimated the effective reproduction number ($R_e$), in each city, over a city-specific time window early in the epidemic. We use a renewal equation-based approach that estimates latent infections and then map these infections to observed notifications via an incubation period, a report delay and a negative binomial observation model with a day of the week effect[18]. Focusing on the early phase of the pandemic allows us to minimise possible biases coming from factors impacting $R_e$ (in particular non-pharmaceutical interventions (NPIs)), which developed as the pandemic progressed. These include change of ascertainment methods and strategies, the implementation of strong NPIs (e.g. travel bans, school closures and lockdown), the appearance of new variants and ultimately vaccination campaigns. Also, in the first stage we define our exposure variables as mean values of meteorological variables (including daily mean temperature, relative and absolute humidity, solar radiation, wind speed and precipitation), for each city, over the early-phase time window, using the ERA5 fifth-generation European Centre for Medium-Range Weather Forecast atmospheric reanalysis of the global climate[19]. In a second 'cross-sectional' stage, we estimate the association of city-level $R_e$, calculated for the city-specific window (allowing for standard errors), with each meteorological variable, controlling for confounding by total population, population density, gross domestic product (GDP) per capita, percentage of population >65 years, pollution levels (i.e. particulate matter, $PM_{2.5}$) and the lagged Oxford COVID-19 Government Response Tracker (OxCGRT) Government Response Index at the endpoint of the selected time window (lagged by 10 days), allowing for the two-level (cities and countries) structure of the data using a multilevel meta-regression model[20] (see 'Methods' for further details). We believe the data used and the analysis performed in this study improves upon previous approaches. Specifically, the fine spatial scale of the city-level data and the methodological design, accounting for confounding of NPIs and city-level covariates, allows us to accurately quantify the relationship between meteorological variables and $R_e$.

## Results

**Descriptive analysis of meteorological variables and $R_e$.** The bivariate distribution of mean temperature and the effective reproduction number ($R_e$) across the 409 study cities is shown in Fig. 1, and the characteristics of the 26 countries are reported in Table 1. The mean effective reproduction number ($R_e$) across all cities was 1.4, ranging from 0.7 to 2.1, with all but ten cities experiencing an epidemic curve with a reproduction number >1. Mean temperatures over the observation period (between January and April 2020) reflect the late winter/early spring in 381 cities situated in the northern hemisphere and the summer/early autumn seasons in 28 cities in the southern hemisphere. Of the 136 cities classified as having high $R_e$ values, 35 cities experienced low temperatures, 64 medium temperatures and 34 high temperatures (Fig. 1). When visualising the unadjusted association of $R_e$ with mean temperature, relative humidity (RH), absolute humidity (AH), solar radiation at the surface and stratified by climate zone, we found no clear pattern (Fig. 2).

**Associations between meteorological variables and $R_e$.** Using a two-stage meta-regression model, we quantified the influence of meteorological variables, including mean temperature, on $R_e$ between cities, while controlling for confounding factors including government interventions. After adjusting for the city-level characteristics (e.g. socio-economic and demographic factors) and the country's OxCGRT Government Response Index, we found a modest, non-linear association of mean temperature and AH with $R_e$ (Table 2). Less strong evidence of association was found for RH, with no evidence of association for solar radiation, wind speed and precipitation (Table 2). The association between mean temperature and $R_e$ is non-linear, with $R_e$ initially rising to a peak at 10.2 °C, then falling to a trough at 20 °C, 0.087 (95% confidence interval (CI): 0.025; 0.148) lower than the peak, and finally rising again (Fig. 3). AH has a similar non-linear shape with a maximum difference of 0.061 (95% CI: 0.011; 0.111) between the peak at 6.6 g/m³ and the trough at 11 g/m³.

**The effect of NPIs on $R_e$.** Although we calculated $R_e$ over a time window in which the OxCGRT Government Response Index, lagged by 10 days, had not yet reached 70, we included the value of the lagged OxCGRT Government Response Index at the end of the city-specific window in the model, to control for residual confounding. Despite being capped at 70, the OxCGRT Government Response Index had a strong association with the reproduction number ($p < 0.0001$) (Supplementary Table 4), explaining 13.8% of its variability (Fig. 3 and Supplementary Table 4, Models D1–D7) with an estimated reduction of $R_e$ equal to 0.285 (95% CI: 0.223; 0.347) when levels of the Government Response Index increase from 21 (5th percentile) to 66 (95th percentile). Mean temperature explained 2.4% and AH 2.0% of the variation in $R_e$, and the five city-level characteristics explained 1.4% of the variability of the reproduction number (Supplementary Table 4, Models D1–D8).

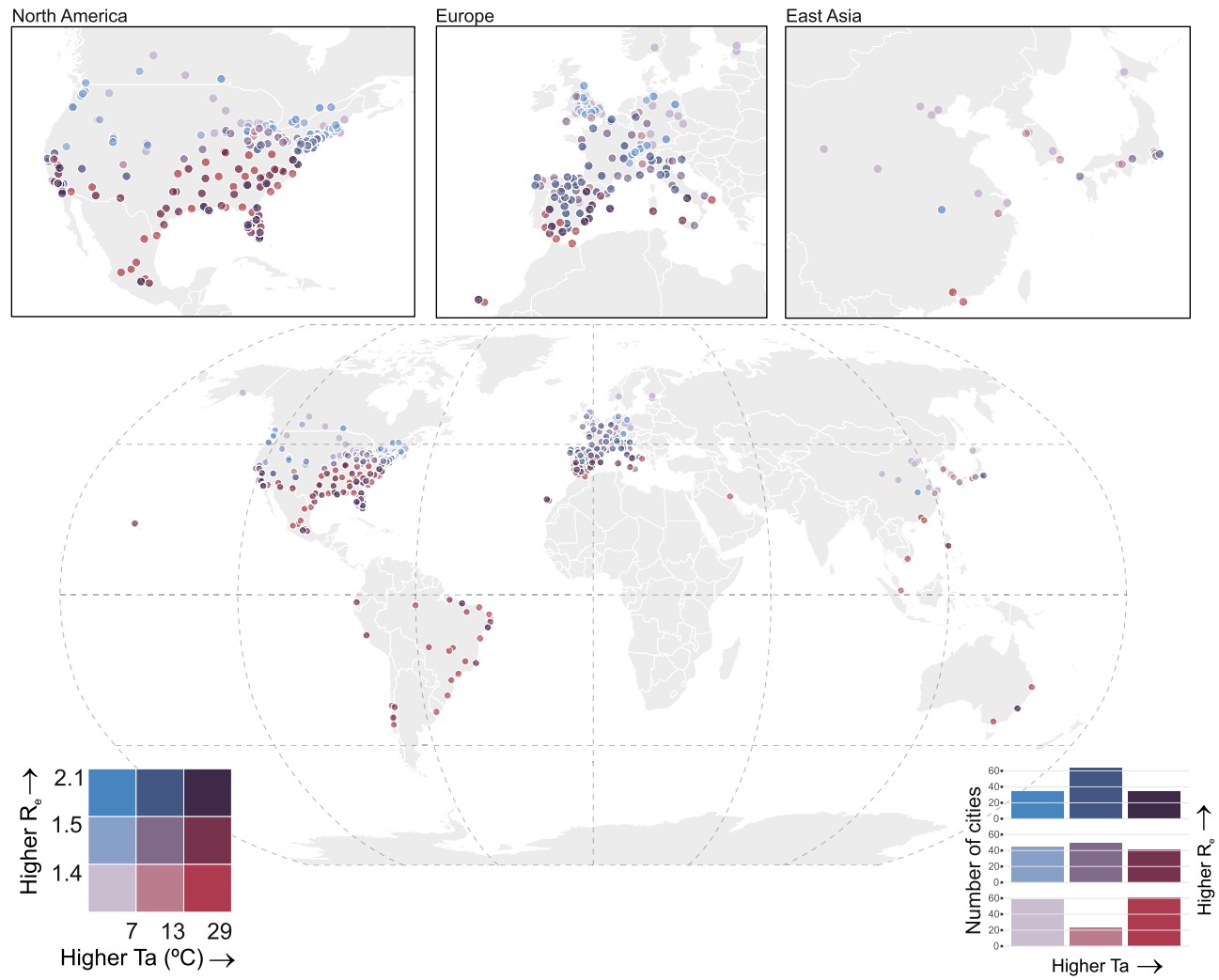

**Fig. 1 Effective reproduction number and mean temperature in the observation window for 409 cities.** Bivariate plot of effective reproduction number ($R_e$) and mean temperature (Ta) (°C) in the observation window for each of the 409 study cities. Dark purple circles represent cities with both high $R_e$ and high Ta, while pale purple circles show areas with both low $R_e$ and low Ta. Red circles represent cities with low $R_e$ and high Ta and blue circles depict areas with high $R_e$ and low Ta. The bar chart (bottom right) represents the number of cities in each category defined in the bivariate legend (bottom left).

**Sensitivity analyses**. We performed several sensitivity analyses to evaluate the robustness of the results considering alternative analytic or selection choices (see Supplementary Table 5). The main results are stable when including a country-level fixed effect in the meta-regression model, i.e. considering the only within-country variation of covariates and outcome. Restricting the analysis to cities with weaker interventions (OxCGRT Government Response Index <60) also gives similar results to the main analysis, apart from wind speed and precipitation also showing an association with $R_e$. The association between mean temperature and the effective reproduction number holds across all the sensitivity analyses, apart from in tropical and southern hemisphere cities, when stratifying by tropical and non-tropical or northern and southern hemisphere regions. However, this may be explained by the small number of cities and the resulting low power in the tropical and southern hemisphere sub-group. The association between AH and the effective reproduction number is somewhat less robust with no association observed when excluding tropical or southern hemisphere cities, when excluding China and Brazil (countries with earlier and later observation periods) and when considering meteorological variables lagged by 10 days. Excluding the ten cities with $R_e < 1$ shows a tendency of an increased $R_e$ for cities with low RH ($p = 0.009$) and a lower $R_e$

in cities with higher solar radiation at the surface ($p = 0.047$) (Supplementary Figure 5). We observed similar overall tendencies to our main results when we did not control for the OxCGRT Government Response Index in our model, although the effect of temperature and AH was enhanced (Supplementary Figure 6), and when considering meteorological variables lagged by 10 days (Supplementary Table 5). We found no evidence of an interaction between mean temperature and RH categorised in two levels (≤65% and >65%) using the median value of 65% as the category threshold ($p = 0.428$).

## Discussion

We combined datasets of COVID-19 transmission with meteorological, demographic, socio-economic and intervention data for 409 cities in 26 countries across the world to estimate the association between meteorological factors and $R_e$ in the early phase of the COVID-19 pandemic. We found evidence of a modest non-monotonic association of outdoor mean temperature and AH with early-phase $R_e$, after controlling for potential confounders, including NPIs. Temperature explained 2.4% and AH 2.0% of the variation in $R_e$, compared to 13.8% explained by the OxCGRT Government Response Index in the adjusted analysis. The associations of temperature and AH with $R_e$ were not

**Table 1 Characteristics of the 26 countries included in the study.**

| Country | Number of cities | Reported COVID-19 cases | Mid-period | $R_e$ | Government index |
|---|---|---|---|---|---|
| Australia | 3 | 1747 | 20/03/2020 | 1.39 | 38.5 |
| Brazil | 18 | 17,179 | 10/04/2020 | 1.29 | 61.9 |
| Canada | 9 | 2709 | 21/03/2020 | 1.50 | 58.9 |
| Chile | 4 | 2587 | 27/03/2020 | 1.32 | 55.9 |
| China | 11 | 4178 | 03/02/2020 | 1.13 | 57.3 |
| Czech Republic | 1 | 358 | 21/03/2020 | 1.36 | 69.2 |
| Ecuador | 1 | 1014 | 20/03/2020 | 1.39 | 46.2 |
| Estonia | 1 | 209 | 20/03/2020 | 1.16 | 41.0 |
| Finland | 1 | 710 | 16/03/2020 | 1.37 | 30.1 |
| France | 17 | 5834 | 17/03/2020 | 1.51 | 55.8 |
| Germany | 12 | 7759 | 16/03/2020 | 1.43 | 41.1 |
| Italy | 19 | 11,796 | 11/03/2020 | 1.49 | 67.9 |
| Japan | 9 | 1178 | 12/03/2020 | 1.29 | 37.0 |
| Kuwait | 1 | 108 | 05/03/2020 | 1.31 | 21.8 |
| Mexico | 8 | 1894 | 25/03/2020 | 1.25 | 28.4 |
| Norway | 1 | 626 | 12/03/2020 | 1.32 | 16.7 |
| Peru | 1 | 428 | 18/03/2020 | 1.45 | 57.7 |
| Philippines | 2 | 215 | 21/03/2020 | 1.40 | 64.7 |
| Singapore | 1 | 56 | 15/02/2020 | 0.87 | 30.1 |
| South Korea | 4 | 5877 | 06/03/2020 | 1.17 | 54.8 |
| Spain | 52 | 43,331 | 11/03/2020 | 1.51 | 42.1 |
| Switzerland | 7 | 6908 | 13/03/2020 | 1.54 | 34.3 |
| United Kingdom | 45 | 9354 | 26/03/2020 | 1.41 | 58.0 |
| United States | 179 | 136,303 | 27/03/2020 | 1.45 | 60.7 |
| Uruguay | 1 | 271 | 19/03/2020 | 0.91 | 46.2 |
| Vietnam | 1 | 38 | 25/03/2020 | 1.10 | 45.5 |

The number of cities per country, total reported COVID-19 cases in the time window, mid-period of the pre-defined window of early transmission, effective reproduction number ($R_e$) and the lagged OxCGRT Government Response Index at the endpoint of the pre-defined window.

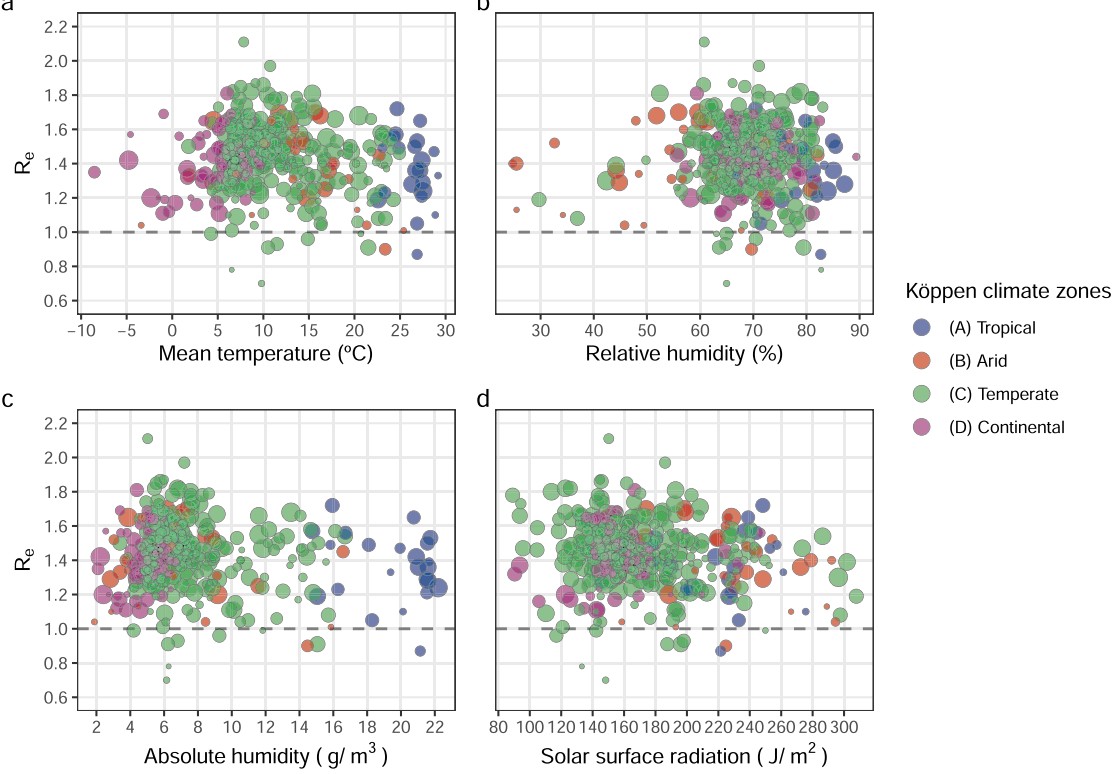

**Fig. 2 Effective reproduction number vs key weather variables by climate zone. a** Mean temperature (°C), **b** relative humidity (%), **c** absolute humidity (g/m³) and **d** solar surface radiation (J/m²) vs effective reproduction number ($R_e$) by climate zone (409 cities). The area of the circles is proportional to the precision (inverse of the variance) of $R_e$ estimates.

**Table 2 Association between weather variables and $R_e$.**

| +Variables | Contrast for which effect size is presented[a] | Effect size 95% CI | P value | Difference in the likelihood ratio $R_{LR}^2$ statistic |
|---|---|---|---|---|
| Mean temperature (°C) | 10.2 vs 20 | 0.087 (0.025; 0.148) | 0.014 | +2.5 |
| Absolute humidity (g/m³) | 6.6 vs 11 | 0.061 (0.011; 0.112) | 0.036 | +2.0 |
| Relative humidity (%) | 60 vs 75 | 0.043 (−0.001; 0.087) | 0.058 | +1.5 |
| Surface solar radiation downwards (J/m²) | 248 vs 124 | −0.053 (−0.117; 0.011) | 0.208 | +0.6 |
| Wind speed (m/s) | 1.1 vs 3.0 | −0.038 (−0.090; 0.014) | 0.152 | +0.7 |
| Total precipitation (m) | 0.1 vs 6 | −0.031 (−0.075; 0.014) | 0.175 | +0.4 |
| OxCGRT (0–100) | 21 vs 66 | 0.285 (0.223; 0.347) | <0.0001 | +13.8 |

Effect size and variation explained by including, in turn, mean temperature (°C), absolute humidity (g/m³), relative humidity (%), surface solar radiation downwards (J/m²), wind speed (m/s), total precipitation (m) and OxCGRT (0–100) in the model of $R_e$. P values were obtained from a two-sided Wald test in the multivariable meta-regression multilevel models adjusted by population (log scale), population density (log scale), GDP (log scale), % population >65 years, PM2.5 (μg/m³, log scale) and the OxCGRT Government Response Index, with cities nested within countries.
[a]The exposure contrast for which effect size is presented is that between the values predicting minimum and maximum $R_e$, where clear minima and maxima are observed (mean temperature, absolute humidity and relative humidity), otherwise the 5th to 95th percentiles.

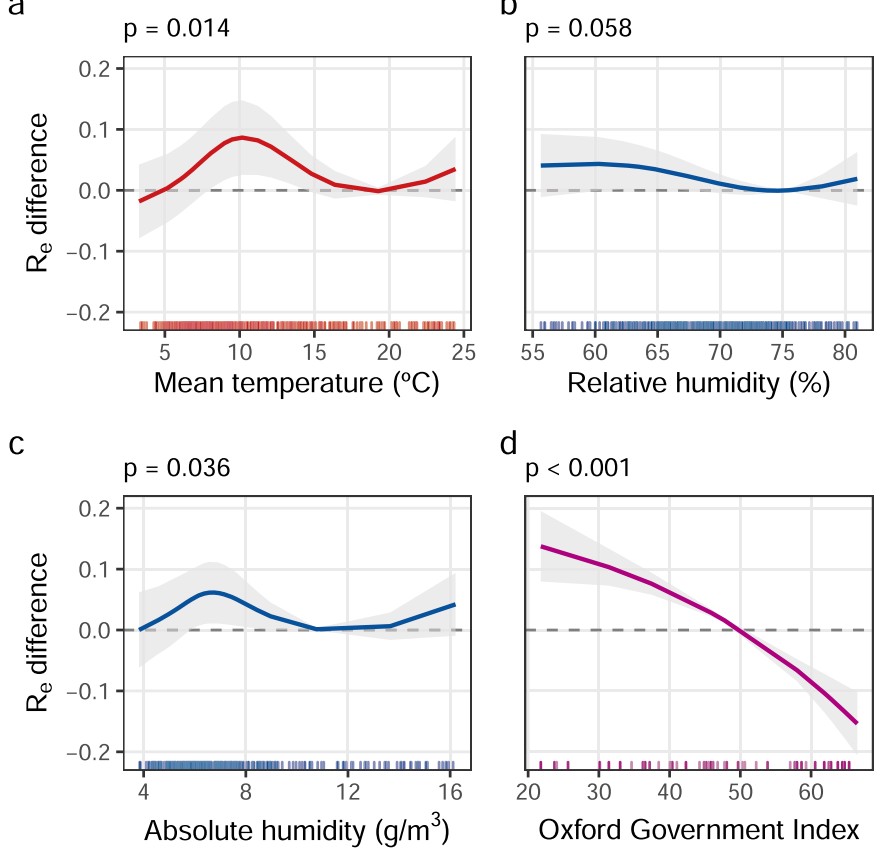

**Fig. 3 Associations between weather variables, non-pharmaceutical interventions and the effective reproduction number.** Non-linear associations between (**a**) mean temperature (°C), (**b**) relative humidity (%), (**c**) absolute humidity (g/m³) and (**d**) OxCGRT Government Response Index and predicted $R_e$ difference. Curves and their 95% confidence intervals show the predicted difference in $R_e$ with respect to a reference value set to the value at the trough of the curve for meteorological variables (**a**–**c**), or for the OxCGRT Government Response Index = 50 (**d**). Two-sided Wald test p values and adjusted curves with 95% confidence intervals were obtained from multivariable meta-regression multilevel models adjusted by population (log scale), population density (log scale), GDP (log scale), % population >65 years of age, PM2.5 (μg/m³, log scale) and OxCGRT Government Response Index, with cities nested within countries. The marginal distribution along the x-axis represents the observed data for that covariate.

independent; the high correlation between them precluded control of one for the other. Overall, there was little evidence for any change in the $R_e$ of COVID-19 associated with RH and no evidence for precipitation and wind speed.

Associations between temperature, humidity and SARS-CoV-2 transmission might be explained by three mechanisms. First, like other viruses with a lipid envelope, SARS-CoV-2 has been found to be sensitive to temperature, humidity and solar radiation under

laboratory conditions[21–25], which affects its ability to survive on surfaces and in aerosols. The droplet behaviour in aerosols changes with different temperature and humidity levels. Low RH promotes the accumulation of aerosol particles (since evaporation leaves behind floating droplet nuclei) increasing the likelihood to be inhaled[26,27]. Second, innate and adaptive immune response mechanisms have been shown to be modulated by seasonal changes. Lower levels of vitamin D, mediated by decreased

ultraviolet B radiation exposure during winter might lead to impaired antiviral innate immune defences[28–30]. Breathing dry air can impair mucociliary clearance, reducing the ability of cilia cells to secrete mucus and remove viral particles (innate immune response)[27,31]. Interferon-stimulated genes, usually inducing an antiviral state as part of the innate immune response have been found to be impaired at low RH[32]. High temperatures have been shown to hinder virus-specific $CD8^+$ T cell responses and antibody production (adaptive immune system)[33]. Third, human mobility, contact patterns and time spent indoors are affected by weather conditions[34]. Very hot and very cold conditions can lead to more time spent in enclosed spaces, which might increase the likelihood of SARS-CoV-2 transmission.

Findings from this study are only partly consistent with findings from other global studies using statistical approaches to investigate meteorological effects on COVID-19 transmission. Meyer et al.[9] found that mean temperature had a modest negative association with COVID-19 incidence for temperatures above $-15\,°C$ based on a dataset of 100 countries, after controlling for surveillance capacity, time since first reported case, population density and median population age, whereas RH had a negative non-significant association with case incidence. Jüni et al.[13] covering 144 geopolitical areas showed that temperature and humidity measures were not significantly associated with epidemic growth while significant associations were found for restrictions of mass gatherings, school closures and measures of social distancing, which are consistent with our findings of a stronger impact of the OxCGRT Government Response Index compared to climatic conditions. Wu et al.[35] incorporating data from 166 countries found that a 1 °C increase in temperature and RH was associated with a 3% and 0.85% decrease in daily new cases, respectively, after controlling for wind speed, median population age, Global Health Index, Human Development Index and population density. Interestingly, non-linear associations between mean daily temperature and the instantaneous reproduction number ($R_t$) in the United States of America were found in a study by Rubin et al.[12] with $R_t$ decreasing to a minimum as temperatures rose to 11 °C, increasing between 11 and 20 °C and then declining again at temperatures >20 °C. The shape of the association may be influenced by the indirect effect of weather in varying the likelihood of social interactions, e.g. at higher temperatures people may congregate in public cities, such as beaches and festivals[12], while colder temperatures could limit social activities, such as sporting events[34]. Runkle et al.[11] concluded from varying longitudinal associations in four cities that specific humidity in the range of 6–9 g/kg (i.e. AH range of 7.6–11.4 g/m³) was a significant predictor of the COVID-19 growth rate, in line with our findings.

Unclear and inconsistent findings related to temperature and humidity may be due to methodological challenges and data limitations. Similar methodological challenges were highlighted when evaluating the association between air pollution and COVID-19 outbreaks[36,37]. The novelty of the virus, with less than a full annual cycle of data available in most places, makes it difficult to disentangle a seasonal signal or inter-annual trends from meteorological factors using time-series models[38]. Moreover, different interventions (e.g. restrictions of mass gatherings, international travel and school closures) adopted by countries at different times after the onset of local outbreaks potentially confound the association between weather variables and COVID-19 spread. These challenges have led us to consider an ecological approach where we compared the outbreak curve early in the epidemic, minimising the confounding effect of NPIs. Despite this, we found a strong association of the OxCGRT Government Response Index with $R_e$, confirming the importance of interventions implemented early on in the epidemic in controlling COVID-19 dynamics[39].

Comparing the early-phase outbreak curves in different countries is challenging given that countries have different case definitions, and early COVID-19 data only captured a small portion of cases, mainly hospitalised patients or individuals with severe symptoms. The estimated high proportion of asymptomatic cases compromises the use of COVID-19 case counts to estimate transmission dynamics[40]. We used an estimated response variable, i.e. the effective reproduction number, calculated accounting for reporting delays and other sources of uncertainty. The 20-day duration was chosen as a compromise between needing enough days for a more precise $R_e$ estimation, while, at the same time, limiting the window to provide more constant weather, case ascertainment and $R_e$ estimates within the window. A larger window would bias estimates in ways that cannot be readily adjusted for. Our meta-analysis approach accounts for the uncertainty in $R_e$ estimates, which in turn reduces the level of certainty in the results. Further, 20 days is ~4 generations of infections, which, under most reporting scenarios, is sufficient to be confident about estimates in the level of transmission. We assume that within the 20-day time window, the case definition is constant within a city or country and $R_e$ is not affected by differences in case definition between countries.

A clear strength of this study is the use of an extensive dataset of 409 cities, representing 44.8% of all cumulative reported COVID-19 cases registered by 31 May 2020 in the John Hopkins University Coronavirus Resource Center. Our analysis covers all major climate zones across the globe, ranging from temperate, continental to tropical and dry climate settings. Another strength is our flexible methodological and statistical approach. We used multilevel meta-analytic models that take into account uncertainty of the response variable, i.e. the effective reproduction number. The model allowed for possible non-linearity of the exposures, and we adjusted for a selection of key socio-economic and demographic factors, as well as using a random effect to account for the country- and city-level differences. We chose covariates based on their potential impact on viral transmission that might confound the examined association of weather and COVID-19 dynamics. Indeed, population density leads to higher contact rates, potentially increasing the likelihood of transmission[41]. The age structure of a population is relevant given that elderly people were found to be more susceptible to infection and more likely to experience clinical symptoms of COVID-19 compared to younger age groups, increasing the likelihood of seeking medical care and getting tested[42]. Moreover, differences in contact patterns among different age groups can further affect the number of COVID-19 cases in each age group[42]. Socio-economic indicators, such as GDP per capita, are important to consider as more deprived populations might be at higher risk of infection due to potential conditions of overcrowded accommodation, inability to work from home or limited access to medical care[43]. Also, among air quality factors, a positive association between $PM_{2.5}$ and COVID-19 incidence and mortality has been reported[44,45].

We investigated model uncertainties with several sensitivity analyses, e.g. excluding cities with $R < 1$, excluding China and Brazil, cities in the southern hemisphere, cities with a latitude lower than 45°, and cities with an OxCGRT Government Response Index of more than 60. Previous studies compared cities within a country or considered large geographical units[13,35,46], which could lead to a limited exposure range with narrow temperature and humidity variability reported during winter seasons, or high measurement errors for meteorological variables defined over large geographical units. We considered small area/city units distributed among 26 countries worldwide, allowing a good exposure range and minimising the measurement error of the exposures.

Our study has several important limitations in addition to those already discussed. Cities in the northern hemisphere were overrepresented compared to southern hemisphere cities, which indicates that the findings might be more representative for cities in the global north. Our results need to be put into the context of complex uncertainties surrounding characteristics of the novel virus, such as incomplete knowledge on possible underlying mechanisms between weather conditions and the virus itself, the role of host immunity and the potential influence of weather-sensitive human behaviour, such as indoor crowding[47]. However, AH was found to demonstrate the strongest indoor-to-outdoor correlation, indicating that outdoor AH measures could reflect indoor conditions[48,49]. Data limitations regarding the novel virus, including varying accuracy of COVID-19 case numbers, limited data availability across cities and temporal constraints of an incomplete seasonal cycle of SARS-CoV-2 contribute to the limitations of this analysis.

Despite these limitations, the associations of weather with $R_e$ in this study suggests that such effects are likely to be small compared to other drivers of transmission. NPIs had a stronger impact on variation in transmission between cities than meteorological variables. We found no weather conditions in which transmission is impeded if precautions (social distancing, mask use, etc.) are not taken. These results support the statement that, to date, COVID-19 interventions are critical regardless of meteorological conditions.

## Methods

**Data**. Data in this study were obtained from a well-established MCC Collaborative Research Network[50]. The current MCC network covers 750 locations (cities or regions) in 43 countries/regions. For this study, 26 countries provided a daily time series of COVID-19 cases for a total of 502 locations (cities or small regions). COVID-19 data were downloaded from a publicly available repository or obtained from health agencies (Supplementary Table 1) and data management was performed using Microsoft Excel 2019. The time series from 1 January 2020 to 31 May 2020 comprises 2,771,137 COVID-19 cases, representing 44.8% of the cumulative cases registered by 31 May 2020 in the Johns Hopkins database (https://coronavirus.jhu.edu/). Supplementary Table 1 shows the sources used for each country along with the definition of COVID-19 cases.

To limit potential confounding by NPIs and temporal variation in case ascertainment, we selected a 10–20-day window early in the epidemic, starting after at least ten confirmed cases had occurred in a 10-day period, to reduce noise introduced via imported cases. We excluded days for which the OxCGRT Government Response Index exceeded 70, accepting reduced windows down to 10 days in length. The OxCGRT collates publicly available information on 18 indicators about governments' policy responses to the COVID-19 pandemic (https://www.bsg.ox.ac.uk/research/research-projects/coronavirus-government-response-tracker). These indicators are categorised as containment or closure policies (e.g. school and workplace closures, restrictions on gatherings and movement), economic policies (e.g. income support) or health policies (e.g. COVID-19 testing programmes). The OxCGRT Government Response Index aggregates these indicators into a single score between 0 and 100 and provides a measure of how many policies a government has enacted, and to what degree. We chose 70 as the maximum value of OxCGRT Government Response Index allowable as a compromise between limiting confounding by government interventions and including enough cities to enable estimation of the associations studied (see Supplementary text 1). Applying these conditions/restrictions reduced our dataset to 409 cities or small regions in 26 countries with an observation period between 11 January 2020 and 28 April 2020.

Most of the 409 cities are situated in the northern hemisphere ($n = 381$), and in temperate (n = 292) or continental (n = 65) climatic zones, with few cities located in tropical (n = 23) and dry (n = 29) climatic zones. The COVID-19 cases were observed in the early phase of the epidemics, ranging from the first week of February 2020 in China to mid-April 2020 in Brazil (Supplementary Figure 1). This early epidemic phase is characterised in many countries (except Uruguay and Singapore) by a reproduction number >1 (Table 1).

We estimated $R_e$ for infections in the time window of interest using EpiNow2 1.3.2[18]. This R package implements a Bayesian latent variable method for estimating $R_e$, where infections at time $t$ are estimated using the sum of previous infections, weighted by an uncertain, gamma-distributed, generation time probability mass function, and multiplied by an estimate of $R_e$[51,52]. Initial infections (prior to the first reported case) were estimated using a log-linear model with priors based on the observed growth in cases. Complete infection trajectories were then mapped to reported case counts by first convolving over the incubation

period distribution and an estimated distribution representing the delay between symptom onset and case report (both assumed to be log-normal). Reporting noise was then added using a negative binomial observation model combined with a multiplicative day of the week effect (modelled using a simplex). $R_e$ was considered to be piecewise constant with a breakpoint 3 days into the time window. The $R_e$ estimate from the first 3 days of the window was discarded and the $R_e$ estimate from the remainder of the window used in all analyses. Each region was fitted independently using Markov chain Monte Carlo. Four chains were used with a warmup of 1000 samples and 4000 samples post warmup. Convergence was assessed using the R hat diagnostic[53].

We used a gamma-distributed generation time with a mean of 3.6 days (standard deviation (SD) 0.7) and a SD of 3.1 days (SD 0.8)[54,55]. This generation time was slightly shorter than the consensus estimate reported by Ferretti et al.[56], leading to our $R_e$ estimates and subsequent effect sizes being conservative. We used a log-normally distributed prior for the incubation period with a mean of 5.2 days (SD 1.1) and SD of 1.52 days (SD 1.1)[57]. The log-normal prior for the delay from symptom onset to case report was estimated globally using a subsampled Bayesian bootstrapping approach (with 100 subsamples each using 250 samples) using data from an international line list of cases. The resulting distribution had a mean of 6.4 days and a standard deviation (SD) of 17 days (or a log mean of 0.83 (SD 0.15) and a log SD 1.44 (SD 0.12). The subsampled bootstrap approach was used to incorporate both the temporal and spatial uncertainty in the reporting distribution as data specific to each setting and time point was not available.

To define our exposures, we considered the following time series from the ERA5 dataset: 2 m temperature, 2 m dewpoint temperature, surface solar radiation downwards, precipitation, and 10 m eastward ($u$) and northward ($v$) components of wind. These are published by the Copernicus Climate Change service on a regular latitude/longitude grid of 0.25° (~25 km × 25 km) in NetCDF format[19]. The hourly 2 m temperature, 2 m dewpoint temperature and surface solar radiation downwards were averaged for each day to derive daily mean temperature, dewpoint temperature and surface solar radiation. From dewpoint temperature and the corresponding temperature ($T$; °C) we obtained RH (%) using the R 'humidity' 0.1.5 package[58]. The following formula was used to calculate AH (g/m³), which represents the mass of water vapour in the air mixture[59]:

$$\mathrm{AH} = (6.112 \times e^{(17.67 \times T)/(T+243.5)} \times 2.1674 \times \mathrm{RH})/(273.15 + T).$$

The hourly 10 m $u$ and $v$ components of wind were averaged for each day, and the daily average $u$ and $v$ components were used to compute the wind speed using the formula wind speed = sqrt($u^2 + v^2$). Hourly precipitation data were summed to derive daily totals. The daily variables were calculated for each 25 km² grid cell and assigned to a city if the city centroid fell within the grid cell.

Mean temperature (and other meteorological variables, Supplementary Table 3) observed during the city-specific time window reflect the late winter/early spring observation period in cities situated in the northern hemisphere and in temperate or continental climatic zones. We found a high correlation between mean temperature and AH (Supplementary Figure 2). Socio-economic and demographic characteristics were extracted from the OECD Regional and Metropolitan database[60] and Worldcities database[61] (Supplementary Table 2). We selected, a priori, the following set of confounders: total population, population density, % elderly population (>65 years) and GDP (per capita). Pollution data (PM₂.₅) for the observation period (10–20 days) was obtained from the Copernicus Atmosphere Monitoring Service global near-real-time service[62–64]. This product provides hourly modelled values of surface PM₂.₅ (μg/m³) at a 0.4 × 0.4 arc degrees grid cell resolution. The hourly time series were averaged over the observation period and linked to the city using the city centroid coordinates. Cities vary in terms of socio-demographic characteristics (Supplementary Table 3). The correlation between socio-demographic characteristics is shown in Supplementary Figure 3 and the correlation between meteorological variables, OxCGRT Government Response Index, day of the year and $R_e$ in Supplementary Figure 4. To account for differences in NPIs we used the OxCGRT Government Response Index[65]. In this study, we considered the 10 days lagged value of the OxCGRT Government Response Index, and for each city, we assigned the index on the last day of the specified window for each city[39]. Note, in our analysis, meteorological variables and socio-demographic covariates were collated and summarised at the city level, while the COVID-19 time series were defined at the smallest administrative level containing the city. We only included cities for which the COVID-19 time series were available for an area in which most of the population resided in that city. We, therefore, refer to our unit of analysis as a city.

**Statistical analysis**. For descriptive purposes, the following statistics (mean, standard deviation and range) were calculated for meteorological variables (mean temperature, AH, RH, surface solar radiation, wind speed, total precipitation) and covariates considered in this study (total population, population density, % elderly population (>65 years), GDP (per capita), PM₂.₅, OxCGRT Government Response Index). We also calculated the correlation (Pearson coefficient) among meteorological variables and among covariates.

The association between city-level covariates and climatic variables with the effective reproduction number were evaluated using multilevel meta-regression models with two levels (cities nested within countries)[20] using the R 'mixmeta' 1.1.0 package. The inclusion of country as a random effect allowed the model to

account for country differences (e.g. data reporting) with efficient use of the within- and between-country information. Moreover, the meta-regression models allowed us to consider the precision of the $R_e$ estimates, as estimated by its variance, giving less weight to more imprecise estimates for shorter time windows.

Firstly, we used two-level meta-regression models to evaluate the possible non-linear association between each meteorological variable and the reproduction number $R_e$. We considered possible non-linearity in the association with $R_e$ using a natural spline parameterisation of the meteorological variables with a variable number of internal knots from 0 (linear term) to 5, placed at respective percentiles of the variable. We compare the models with different non-linear parameterisations of the meteorological variable using the Akaike Information Criteria (AIC), choosing models with the lowest AIC.

We fitted the following two-level random-effects meta-regression models with cities nested within countries and an increasing number of predictors; Model A with two random effects (cities and countries) and the intercepts, Model B including the OxCGRT Government Response Index, Model C considering also total population, density, GDP, % population older than 65 years, and PM$_{2.5}$ (total population, density, GDP and PM$_{2.5}$ were log-transformed due to the skewness of their distribution).

Then for each meteorological variable, we fitted two-level meta-regression models (D1–D6) with the meteorological variable as exposure and total population, density, GDP, % population older than 65 years, PM$_{2.5}$ and the OxCGRT Government Response Index as covariates. We considered non-linearity in the association with $R_e$ using a natural spline parameterisation of the climatic variables with the number of internal knots as determined in the univariate analysis. The coefficients of the natural spline parameterisation of the meteorological variable were used to derive the plot of the association between the meteorological variable and $R_e$ in the 5–95th percentile of the meteorological variable distribution (Fig. 3 and Supplementary Figures 5 and 6). The coefficients of the natural spline parameterisation of the meteorological variable were also used to test the association between the meteorological variable and $R_e$ using the multivariate Wald test. All the tests were two-sided. Given the small number of pre-defined exposures variables, no adjustment was made for multiple comparisons.

We quantified heterogeneity between cities with standard measures of $I^2$ [66]. These measures are estimated once from a meta-regression model without meta-predictors (Model A) and once from the meta-regression models (Models B, C and D1–D6) to assess the residual heterogeneity provided by the different set of predictors. For each model, we calculated the likelihood ratio test ($R_{LR}^2$) statistic[67]. $R_{LR}^2$ is calculated as $1 - \exp(-2/409 \times (\log \text{Lik}_m - \log \text{Lik}_0))$, where $\log \text{Lik}_m$ is the log-likelihood of the model of interest and $\log \text{Lik}_0$ is the log-likelihood from a null model including only city and country random effect (i.e. Model A). For each meteorological variable, we calculated the difference in the likelihood ratio test $R^2$ ($R_{LR}^2$) with respect to Model C (including random effects, OxCGRT Government Response Index and city-level covariates). For OxCGRT and city-level covariates, the $R_{LR}^2$ represents the reduction in $R_{LR}^2$ when dropping OxCGRT or city-level covariates from Model D1 with temperature and all other terms (i.e. random effects, OxCGRT and city-level covariates).

**Reporting summary**. Further information on research design is available in the Nature Research Reporting Summary linked to this article.

## Data availability

COVID-19 data were downloaded from publicly available repositories or obtained from health agencies (Supplementary Table 1). COVID-19 data for Australia, Brazil, Canada, Chile, China, Czech Republic, Estonia, Finland, Germany, Italy, Kuwait, Mexico, Norway, Peru, Philippines, Romania, South Korea, Spain, United Kingdom, United States and Vietnam are publicly available. COVID-19 data for Japan and Singapore are available upon request. COVID-19 Data for France, Switzerland and Uruguay were obtained by a specific request to health agencies and are not publicly available.

Meteorological variables (mean temperature, dewpoint temperature, solar radiation, wind components and precipitation) were derived from ERA5 reanalysis product 'https://cds.climate.copernicus.eu/cdsapp#!/search?type=dataset'.

Pollution levels (PM2.5) was derived from CAMS near real time 'https://apps.ecmwf.int/datasets/data/cams-nrealtime/levtype=sfc/'.

The OxCGRT Government Response Index was downloaded from the public repository: https://github.com/OxCGRT/covid-policy-tracker/raw/master/data/OxCGRT_latest.csv (downloaded 3 August 2020).

Socio-economic and demographic characteristics were extracted from the OECD Regional and Metropolitan database 'https://www.oecd.org/regional/regional-policy/regionalstatisticsandindicators.htm' and Worldcities database.

Data were processed and harmonised at the city level. The city-level data used in the main and supplementary analysis of the paper are available in the GitHub directory: https://github.com/fsera/COVIDWeather/ [68].

## Code availability

The code developed in the study to perform the city-level main analysis is available in the following GitHub repository[68].

For each meteorological variable, the effect size was calculated using predicted curves from multivariable meta-regression multilevel models. We calculated the difference in the likelihood ratio test $R^2$ ($R_{LR}^2$) with respect to a model including random effects, OxCGRT Government Response Index, and city-level covariates (Model C, Supplementary Table 4). $R_{LR}^2$ is calculated as $1 - \exp(-2/409 \times (\log \text{Lik}_m - \log \text{Lik}_0))$, where $\log \text{Lik}_m$ is the log-likelihood of the model of interest and $\log \text{Lik}_0$ is the log-likelihood from a null model including only city and country random effect (i.e., Model A, Supplementary Table 4). For OxCGRT, the $R_{LR}^2$ represents the reduction in $R_{LR}^2$ when dropping OxCGRT from the model with temperature and all other terms (i.e., random-effects and city-level covariates).

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

## Acknowledgements

This work was generated using Copernicus Climate Change Service (C3S) and Copernicus Atmosphere Monitoring Service (CAMS) information [2020]. The authors would like to thank the European Centre for Medium-Range Weather Forecasts (ECMWF) that implements the C3S and CAMS on behalf of the European Union. D.R. was supported by a postdoctoral research fellowship of the Xunta de Galicia (Spain). A.G. was funded by the Medical Research Council-UK (Grant ID: MR/R013349/1), the Natural Environment Research Council UK (Grant ID: NE/R009384/1) and the European Union's Horizon 2020 Project Exhaustion (Grant ID: 820655). R.L. was supported by a Royal Society Dorothy Hodgkin Fellowship. S.A. and S.M. were funded by the Wellcome Trust (grant 210758/Z/18/Z210758/Z/18/Z). The following funding sources are acknowledged as providing funding for the MCC Collaborative Research Network authors: J.K. and A.U. were supported by the Czech Science Foundation, project 18-22125S. S.T. was supported by the Shanghai Municipal Science and Technology Commission (Grant 18411951600). N.S. is supported by the NIEHS-funded HERCULES Center (P30ES019776). H.K. was supported by the National Research Foundation of Korea (BK21 Center for Integrative Response to Health Disasters, Graduate School of Public Health, Seoul National University). A.S., F.D.R. and S.R. were funded by the European Union's Horizon 2020 Project Exhaustion (Grant ID: 820655). Each member of the CMMID COVID-19 Working Group contributed to processing, cleaning and interpretation of data, interpreted findings, contributed to the manuscript and approved the work for publication. The following funding sources are acknowledged as providing funding for the CMMID COVID-19 working group authors. This research was partly funded by the Bill & Melinda Gates Foundation (INV-001754: M.Q; INV-003174: K.P., M.J., Y.L., J.L.; NTD Modelling Consortium OPP1184344: C.A.B.P., G.M.; OPP1180644: S.R.P.; OPP1183986: E.S.N.). BMGF (OPP1157270: K.E.A.). DFID/Wellcome Trust (Epidemic Preparedness Coronavirus research programme 221303/Z/20/Z: C.A.B.P.). EDCTP2 (RIA2020EF-2983-CSIGN: H.P.G.). ERC Starting Grant (#757699: M.Q.). This project has received funding from the European Union's Horizon 2020 research and innovation programme—project EpiPose (101003688: K.P., M.J., P.K., R.C.B., W.J.E., Y.L.). This research was partly funded by the Global Challenges Research Fund (GCRF) project 'RECAP' managed through RCUK and ESRC (ES/P010873/1: A.G., C.I.J., T.J.).

HDR UK (MR/S003975/1: R.M.E.). MRC (MR/N013638/1: N.R.W.; MR/V027956/1: W.W.). Nakajima Foundation (A.E.). This research was partly funded by the National Institute for Health Research (NIHR) using UK aid from the UK Government to support global health research. The views expressed in this publication are those of the author(s) and not necessarily those of the NIHR or the UK Department of Health and Social Care (16/136/46: B.J.Q.; 16/137/109: B.J.Q., F.Y.S., M.J., Y.L.; Health Protection Research Unit for Immunisation NIHR200929: N.G.D.; Health Protection Research Unit for Modelling Methodology HPRU-2012-10096: T.J.; NIHR200908: R.M.E.; NIHR200929: F.G.S., M.J.; PR-OD-1017-20002: A.R., W.J.E.). Royal Society (Dorothy Hodgkin Fellowship: R.L.; RP\EA\180004: P.K.). UK DHSC/UK Aid/NIHR (PR-OD-1017-20001: H.P.G.). UK MRC (MC_PC_19065—Covid 19: Understanding the dynamics and drivers of the COVID-19 epidemic using real-time outbreak analytics: A.G., N.G.D., R.M.E., S.C., T.J., W.J.E., Y.L.; MR/P014658/1: G.M.K.). Authors of this research receive funding from the UK Public Health Rapid Support Team funded by the United Kingdom Department of Health and Social Care (T.J.). Wellcome Trust (206250/Z/17/Z: A.J.K., T.W.R.; 206471/Z/17/Z: O.B.; 208812/Z/17/Z: S.C.; 210758/Z/18/Z: J.D.M., J.H., N.I.B.; UNS110424: F.K.). No funding (A.M.F., A.S., C.J.V.-A., D.C.T., J.W., K.E.A., Y.-W.D.C.). LSHTM, DHSC/UKRI COVID-19 Rapid Response Initiative (MR/V028456/1: Y.L.). Innovation Fund of the Joint Federal Committee (01VSF18015: F.K.). Foreign, Commonwealth and Development Office/Wellcome Trust (221303/Z/20/Z: M.K.).

## Author contributions

Conceptualisation—ideas; formulation or evolution of overarching research goals and aims: F.S., B.A., S.A., K.O'R., M.H., M.P., A.T., A.M.V.-C., A.G., R.L. Data curation—management activities to annotate (produce metadata), scrub data and maintain research data (including software code, where it is necessary for interpreting the data itself) for initial use and later re-use: F.S., B.A., S.A. and S.M. Formal analysis—application of statistical, mathematical, computational or other formal techniques to analyse or synthesise study data: F.S., B.A. and S.A. Funding acquisition—acquisition of the financial support for the project leading to this publication: AG. Investigation—conducting a research and investigation process, specifically performing the experiments or data/evidence collection: F.S., S.A., S.M., R.L., R.S. and MCC Collaborative Research Network. Methodology—development or design of methodology; creation of models: F.S., B.A., S.A. and A.G. Project administration—management and coordination responsibility for the research activity planning and execution: F.S. and R.L. Resources—provision of study materials, reagents, materials, patients, laboratory samples, animals, instrumentation, computing resources or other analysis tools: A.G. Software—programming, software development; designing computer programmes; implementation of the computer code and supporting algorithms; testing of existing code components: F.S., B.A., S.A. and A.G. Supervision—oversight and leadership responsibility for the research activity planning and execution, including mentorship external to the core team: F.S., B.A. and R.L. Validation—verification, whether as a part of the activity or separate, of the overall replication/reproducibility of results/experiments and other research outputs: F.S. and S.A. Visualisation—preparation, creation and/or presentation of the published work, specifically visualisation/data presentation: F.S., B.A., S.A., R.L. and D.R. Writing—original draft—preparation, creation and/or presentation of the published work, specifically writing the initial draft (including substantive translation): F.S., R.L., B.A. and R.v.B. Writing—review and editing—preparation, creation and/or presentation of the published work by those from the original research group, specifically critical review, commentary or revision—including pre- or post-publication stages: F.S., B.A., S.A., S.M., K.O'R., R.v.B., R.S., D.R., M.H., M.P., A.T., A.M.V.-C., A.G., R.L., MCC Collaborative Research Network and CMMID COVID-19 Working Group.

## Competing interests

The authors declare no competing interests.

## Additional information

## MCC Collaborative Research Network

Wenbiao Hu[19], Shilu Tong[19,20,21,22], Eric Lavigne[23,24], Patricia Matus Correa[25], Xia Meng[26], Haidong Kan[26], Jan Kynčl[27,28], Aleš Urban[29,30], Hans Orru[31], Niilo R. I. Ryti[32,33], Jouni J. K. Jaakkola[32,33], Simon Cauchemez[34], Marco Dallavalle[35], Alexandra Schneider[35], Ariana Zeka[36], Yasushi Honda[37,38], Chris Fook Sheng Ng[11], Barrak Alahmad[39], Shilpa Rao[40], Francesco Di Ruscio[40], Gabriel Carrasco-Escobar[41,42], Xerxes Seposo[11], Iulian Horia Holobâcă[43], Ho Kim[44], Whanhee Lee[44], Carmen Íñiguez[45], Martina S. Ragettli[46,47], Alicia Aleman[48], Valentina Colistro[49], Michelle L. Bell[50], Antonella Zanobetti[39], Joel Schwartz[39], Tran Ngoc Dang[51], Noah Scovronick[52], Micheline de Sousa Zanotti Stagliorio Coélho[53], Magali Hurtado Diaz[54] & Yuzhou Zhang[55,56]

[19]School of Public Health and Social Work, Queensland University of Technology, Brisbane, QLD, Australia. [20]Shanghai Children's Medical Centre, School of Medicine, Shanghai Jiao-Tong University, Shanghai, China. [21]School of Public Health, Institute of Environment and Human Health, Anhui Medical University, Hefei, China. [22]Center for Global Health, School of Public Health, Nanjing Medical University, Nanjing, China. [23]School of Epidemiology and Public Health, Faculty of Medicine, University of Ottawa, Ottawa, ON, Canada. [24]Air Health Science Division, Health Canada, Ottawa, ON, Canada. [25]Department of Public Health, Universidad de los Andes, Santiago, Chile. [26]Key Lab of Public Health Safety of the Ministry of Education and NHC Key Lab of Health Technology Assessment, School of Public Health, Fudan University, Shanghai, China. [27]Department of Infectious Diseases Eepidemiology, National Institute of Public Health, Prague, Czech Republic. [28]Department of Epidemiology and Biostatistics,

Third Faculty of Medicine, Charles University, Prague, Czech Republic. [29]Institute of Atmospheric Physics of the Czech Academy of Sciences, Prague, Czech Republic. [30]Faculty of Environmental Sciences, Czech University of Life Sciences, Prague, Czech Republic. [31] Institute of Family Medicine and Public Health, University of Tartu, Tartu, Estonia. [32]Center for Environmental and Respiratory Health Research (CERH), University of Oulu, Oulu, Finland. [33]Medical Research Center Oulu (MRC Oulu), Oulu University Hospital and University of Oulu, Oulu, Finland. [34]Mathematical Modelling of Infectious Diseases Unit, Institut Pasteur, Paris, France. [35]Institute of Epidemiology, Helmholtz Zentrum München—German Research Center for Environmental Health (GmbH), Neuherberg, Germany. [36]Institute of Environment, Health and Societies, Brunel University London, London, UK. [37]Center for Climate Change Adaptation, National Institute for Environmental Studies, Tsukuba, Japan. [38]Faculty of Health and Sport Sciences, University of Tsukuba, Tsukuba, Japan. [39]Department of Environmental Health, Harvard T.H. Chan School of Public Health, Harvard University, Boston, MA, USA. [40]Norwegian Institute of Public Health, Oslo, Norway. [41]Health Innovation Laboratory, Institute of Tropical Medicine "Alexander von Humboldt", Universidad Peruana Cayetano Heredia, Lima, Peru. [42]Scripps Institution of Oceanography, University of California San Diego, La Jolla, CA, USA. [43]Faculty of Geography, Babes-Bolyai University, Cluj-Napoca, Romania. [44]Department of Public Health Science, Graduate School of Public Health, Institute of Health and Environment, Seoul National University, Seoul, Republic of Korea. [45]Department of Statistics and Computational Research, Universitat de València, València, Spain. [46]Swiss Tropical and Public Health Institute, Basel, Switzerland. [47]University of Basel, Basel, Switzerland. [48]Departament of Preventive and Social Medicine, School of Medicine, Universidad de la República, Montevideo, Uruguay. [49]Departmente of Cuantitative Methods, School of Medicine, Universidad de la República, Montevideo, Uruguay. [50]School of the Environment, Yale University, New Haven, CT, USA. [51]Department of Environmental Health, Faculty of Public Health, University of Medicine and Pharmacy at Ho Chi Minh City, Ho Chi Minh City, Vietnam. [52]Gangarosa Department of Environmental Health. Rollins School of Public Health, Emory University, Atlanta, GA, USA. [53]Department of Pathology, Faculty of Medicine, University of São Paulo, São Paulo, Brazil. [54]Department of Environmental Health, National Institute of Public Health, Cuernavaca Morelos, Mexico. [55]College of Computer Science and Technology, Zhejiang University, Hangzhou, China. [56]Department of Research, Baolue Technology (Zhejiang) Co., Ltd, Hangzhou, China.

## CMMID COVID-19 Working Group

Timothy W. Russell[3], Mihaly Koltai[3], Adam J. Kucharski[3], Rosanna C. Barnard[3], Matthew Quaife[3], Christopher I. Jarvis[3], Jiayao Lei[3], James D. Munday[3], Yung-Wai Desmond Chan[3], Billy J. Quilty[3], Rosalind M. Eggo[3], Stefan Flasche[3], Anna M. Foss[3], Samuel Clifford[3], Damien C. Tully[3], W. John Edmunds[3], Petra Klepac[3], Oliver Brady[3], Fabienne Krauer[3], Simon R. Procter[3], Thibaut Jombart[3], Alicia Rosello[3], Alicia Showering[3], Sebastian Funk[3], Joel Hellewell[3], Fiona Yueqian Sun[3], Akira Endo[3], Jack Williams[3], Amy Gimma[3], Naomi R. Waterlow[3], Kiesha Prem[3], Nikos I. Bosse[3], Hamish P. Gibbs[3], Katherine E. Atkins[3], Carl A. B. Pearson[3], Yalda Jafari[3], C. Julian Villabona-Arenas[3], Mark Jit[3], Emily S. Nightingale[3], Nicholas G. Davies[3], Kevin van Zandvoort[3], Yang Liu[3], Frank G. Sandmann[3], William Waites[3], Kaja Abbas[3], Graham Medley[3] & Gwenan M. Knight[3]

