## [Peer Review File · Nature Communications]

REVIEWER COMMENTS

Reviewer #1 (Remarks to the Author):

First, I apologize to the authors for my tardiness with this review. Here, Sera and colleagues present a meta-regression model to test for associations between weather covariates and COVID cases. They find small effects for temperature and relative humidity, though these are dominated by a strong effect of NPIs. While the paper is very well written, the figures nice, and the number of cities extensive, I find that the conclusions drawn here are not based on enough data to rule in or out effects of seasonality/weather on COVID cases.

Major points:

- It's not clear why such early dates were chosen, and whether there was enough time for NPIs to be implemented let alone study their effectiveness.

- Similarly, the date ranges used for study (Fig S1 and Table S3) are quite small with most cities having 20 or less days of observation. This seems entirely too short to draw any sorts of conclusions of the effects of weather (or NPIs for that matter) on COVID cases. I think much more data are needed for proper analysis and support for the conclusions drawn, especially for a venue such as Nature Communications.

Minor points:

- The data should be CSV or Excel file, not a table in the supplemental doc.

- Some of the references are incomplete, please double check them.

Reviewer #2 (Remarks to the Author):

Review of Espinosa et al:

Sera et al use short-term data from over 400 cities to test for seasonality in Covid-19 transmission patterns. Their main conclusion that meteorological factors are not important determinants of early local epidemics, esp. when compared to the impact of government interventions and human population behavior. This is in agreement with other observations worldwide that have run counter to earlier predictions that, like other corona viruses, Covid-19 will show a seasonal pattern, favoring winter conditions. This study has the advantage of considering these other key modulating factors.

My only question is regarding the short time period considered, and the impact of timing of arrival of Covid-19 in the different cities. While government response and population behavior were considered, the timing and size of the first wave varied highly between cities, even those within the same climatic zones, at least in part, thanks to preventive measures and preparations taken by cities that were impacted later. Was timing of 10-20 window an explanatory factor?

Given that for most cities, we are now more than a year since the first wave, consideration of the transmission and disease patterns over a longer period of time and/or over several time periods appears to have been possible. While it is true that the less than a full year of data limit analysis of inter-annual patterns, consideration of the relationship between when outbreak window considered and level of preparedness and response may be called for.

Another concern, acknowledged by the authors is that most cities are in the northern hemisphere, with 333 cities from 7 countries and 179 from the U.S. a separate analysis for the few southern countries, and for countries that cover a large range of latitudes may be of interest. Italy, in particular with very different initial outbreak patterns may be especially informative, in addition to the U.S.

With regard to mode of transmission, a consensus appear to have emerged that contact transmission is not an important mechanism, and the impact of meteorological conditions on contact transmission is probably of limited importance for Covid-19 transmission.

Overall, the paper provides a valuable addition to our understanding of the (limited) impact of weather on Covid-19. The description of the data and the methods used is clear and justified, and the tables and figures are effective. The authors provide good discussion of many of the limitations of their analysis, and the paper is timely, of value, and of interest to readers of this journal.

Reviewer #3 (Remarks to the Author):

This is the most detailed analysis of the relation between climate and the early spread of COVID-19 to date. The authors have curated a huge dataset of demographic, epidemiological and climatic variables for over 500 local authorities in 26 countries and on 5 continents. Crucially, the analysis account for local variations in reporting and non-pharmaceutical interventions. Given all the issues and limitations of COVID reporting during the first wave of the pandemic, I think the authors have done a very good job, and it seems very unlikely that they could have substantially underestimated the impact of climatic factors.

I don't have any major issues, so this review is more about dotting the i's and crossing the t's.

There are two obvious gaps in the sampling of the variables. First, large parts of the world are missing, including Africa, Eastern Europe, the Middle East and South Asia. This presumably stems from missing or unreliable case reports in those regions for the first 4 months of 2020. Looking at Figure 2, the available locations appear to cover a very wide range of climatic values, so it's unlikely that large effects of climate on COVID spread could have been missed. The fact that temperate regions (especially Western Europe and the USA) have been much more intensively sampled might hide some trends in the tails of the temperature and humidity distributions. However, I assume that the inclusion of countries as random factors in the statistical model should help rebalance the data. If anything, the over-sampling in temperate regions might have inflated the association between intermediate temperatures and higher R_e .

Second, the authors only considered the initial spread of COVID in each of the locations covered, from January to April 2020. Obviously, there are lots of good reasons to restrict the analysis to the initial period, before sociopolitical factors became too tangled up. However, this may have excluded some countries that reported their first waves later: have you checked?

In theory, it would have been interesting to analyse seasonal variations within countries over several months, but I'm pretty sure the data would be too noisy and the models too complex.

Other questions and comments:

- Although the title and main text only refers to locations as "cities", the locales appear to be a mix of local authorities of various sizes, (e.g. whole départements in France) as briefly acknowledged in the MMethods (l.285). Although I can't think of any obvious issue with the analysis, the use of the more restrictive term "cities" across the manuscript could imply a demographic bias in the selection of data.

- I don't understand why the authors capped the OxGRT index at all: why would values above 70 introduce substantial "confounding by government interventions"? Why is that a confounding factor if it's included in your model?

- Have the authors considered modelling the effects of OxGRT and climatic variables with a time lag? The values of R_e in the short time windows may reflect variations that occurred 2-3 weeks earlier.

- The estimation of R_e was based on consensus distributions for the generation time and incubation periods. Is there any indication that these distributions may differ among countries, e.g. because of demographic differences?

Response to editor and reviewer comments

Manuscript reference number: NCOMMS-21-06776

Title: Potential drivers of COVID-19 seasonality in 409 cities across 26 countries

We wish to thank the editors and reviewers for their valuable comments on our manuscript. We have carefully addressed all the comments (in italics) as detailed below.

Reviewers' comments:

Reviewer #1:

First, I apologize to the authors for my tardiness with this review. Here, Sera and colleagues present a meta-regression model to test for associations between weather covariates and COVID cases. They find small effects for temperature and relative humidity, though these are dominated by a strong effect of NPIs. While the paper is very well written, the figures nice, and the number of cities extensive, I find that the conclusions drawn here are not based on enough data to rule in or out effects of seasonality/weather on COVID cases.

Thank you for the positive feedback, we hope our responses below clarify our choice of study design and justify our conclusions.

Major points:

R1.1. It's not clear why such early dates were chosen, and whether there was enough time for NPIs to be implemented let alone study their effectiveness.

Investigating impacts of NPIs was not the objective of the study. Our aim was to quantify the association between meteorological variables and early COVID-19 transmission dynamics, by limiting and controlling for the impact of government interventions. We considered NPIs (through the OxGGRT index) because of the possibility that government interventions confounded (biased) the estimate of impact of weather on COVID-19. For this purpose, we deliberately excluded days for which the OxCGRT index was high (≥ 70) and included OxCGRT (i.e., we assigned the value of the index, lagged by 10 days, on the last day of the specified window for each city) in the regression analyses, to control for residual confounding. Despite the exclusion of high OxCGRT days, OxCGRT was found to be a strong predictor of the effective reproduction number R_e . We thought it was important to adjust for this and to document the finding, although this was contextual information rather than the focus of the study.

To better clarify this, we have added more explicit aims in the Introductions as follows:

Line 64

“In this study, we overcome methodological issues of previous approaches by using a two-stage ecological modelling approach to examine the impact of meteorological variables on SARS-CoV-2 transmission between cities located across the globe, while accounting for confounding of non-pharmaceutical interventions and city-level covariates.”

Line 72

“To avoid the possibility of non-pharmaceutical interventions (NPIs) confounding the estimates of the impact of weather on COVID-19 transmission dynamics, we defined a city-specific time window (between 10-20 days) for which local transmission has been established but before NPIs had intensified. We first estimated the effective reproduction number (R_e) in each city over the time window early in the epidemic using a renewal equation based approach that estimated latent infections and then mapped these infections to observed notifications via an incubation period, a report delay, and a negative binomial observation model with a day of the week effect¹⁹. The window started after at least 10 cases had occurred in a 10-day period (to reduce bias from imported cases). We only included days where the Oxford COVID-19 Government Response Tracker (OxCGRT) Government Response Index²⁰, lagged by 10 days, was less than 70 (on a scale of 0–100), leaving a total of 409 cities for further analysis. The OxCGRT Government Response Index aggregates 18 indicators about governments' policy responses to the COVID-19 pandemic into a single score, between 0 and 100, and provides a measure of how many policies a government has enacted, and to what degree (with a higher score

indicating a stronger policy response). Focusing on the early phase of the pandemic allowed us to minimise possible biases coming from factors impacting on R_e (in particular NPIs), which developed as the pandemic progressed. These included change of ascertainment methods and strategies, the implementation of strong NPIs (e.g., travel bans, school closures and lockdown) the appearance of new variants, and ultimately vaccination campaigns.”

R1.2. Similarly, the date ranges used for study (Fig S1 and Table S3) are quite small with most cities having 20 or less days of observation. This seems entirely too short to draw any sorts of conclusions of the effects of weather (or NPIs for that matter) on COVID cases. I think much more data are needed for proper analysis and support for the conclusions drawn, especially for a venue such as Nature Communications.

First, it was not our objective to investigate how changes in weather in each city impacted on changes in the COVID-19 effective reproduction number (R_e) in the same city. Rather, we sought to investigate the association between weather and R_e across cities. In epidemiological terms, we were undertaking a cross sectional (or ecological) analysis, not a longitudinal analysis. Given this objective, short time windows had the advantage that weather, case ascertainment, and R_e would be relatively constant within each time window and city.

We were aware that a too short window would make estimation of R_e less precise. In selecting our time window, the primary consideration was finding a window that was as unbiased by case importation, NPIs, variation in case ascertainment, and variation in weather as possible. Whilst we acknowledge that a larger window would have reduced uncertainty in R_e estimates we argue that this reduction in uncertainty would be spurious given the likely introduction of bias from the confounders outlined above. Our use of a meta-analysis modelling approach allowed us to control for uncertainty in R_e estimates though we note that as 20 days is approximately 4 generations of infection it is likely that our estimates do contain sufficient information to provide reasonably accurate estimates of R_e once local transmission was dominant but before NPIs and other confounders dominated estimates. We also note that we used an uninformed prior for R_e meaning that our estimates are not dominated by prior assumptions.

Focusing on the early phase of the pandemic allowed us to minimise possible biases coming from factors impacting on R_e (in particular NPIs), which developed as the pandemic progressed. These included change of ascertainment methods and strategies, the implementation of strong government interventions (e.g., travel bans, school closures and lockdown) the appearance of new variants, and ultimately vaccination campaigns.

Despite our attempt to exclude substantially impactful NPIs (government interventions), it was clear from exploratory analysis that to obtain a sufficient number of cities we would have to include some days by which time NPIs had begun but were limited (e.g. measures like work closures, limits on events/gatherings and international travel controls had been implemented but before stricter measures like closing public transport, school closures and full lockdown had been put in place). Our compromise was to exclude days by which time the OxGGRT index had reached 70. This left some cities with less than 20 eligible days. We included cities providing at least 10 days meeting this criterion. The lower precision in R_e estimates in these shorter windows was taken into account by the reduced weight given to those cities in our cross-sectional mixed effects meta-regression models.

We have edited the text to highlight the study design and justify the choice of window (see response to R1.1. and additions to the discussion below)

Line 247

“The 20-day duration was chosen as a compromise between needing enough days for a more precise R_e estimation while, at the same time, limiting the window to provide more constant weather, case ascertainment, and R_e estimates within the window. A larger window would bias estimates in ways that cannot be readily adjusted for. Our meta-analysis approach accounts for the uncertainty in R_e estimates, which in turn reduces the level of certainty in the results. Further, 20 days is approximately 4 generations of infections, which, under most reporting scenarios, is sufficient to be confident about estimates in the level of transmission.”

Minor points:

R1.3. The data should be CSV or Excel file, not a table in the supplemental doc.

The data are now available as .csv and .RData files in the Github directory:
<https://github.com/fsera/COVIDWeather>

We have removed the data displayed in Table S3 from the supplementary materials and renumbered the subsequent tables accordingly.

R1.4. Some of the references are incomplete, please double check them.

We have checked and updated all references.

Reviewer #2

Sera et al use short-term data from over 400 cities to test for seasonality in Covid-19 transmission patterns. Their main conclusion that meteorological factors are not important determinants of early local epidemics, esp. when compared to the impact of government interventions and human population behavior. This is in agreement with other observations worldwide that have run counter to earlier predictions that, like other corona viruses, Covid-19 will show a seasonal pattern, favoring winter conditions. This study has the advantage of considering these other key modulating factors.

We thank the reviewer for the positive evaluation of our manuscript.

R2.1. My only question is regarding the short time period considered, and the impact of timing of arrival of Covid-19 in the different cities. While government response and population behavior were considered, the timing and size of the first wave varied highly between cities, even those within the same climatic zones, at least in part, thanks to preventive measures and preparations taken by cities that were impacted later. Was timing of 10-20 window an explanatory factor?

We thank the reviewer that highlighted this important point. As shown in the correlation plot below, there was a positive correlation ($r = 0.44$) between day of the year and the OxCGRT index, suggesting a higher level of government intervention in cities that were impacted later. Day of the year was also positively correlated with temperature ($r = 0.27$), which was influenced by the large number of northern hemisphere cities in our database.

[New] Fig. S4. Correlations between meteorological variables (Ta=Air temperature, RH=Relative humidity, AH=Absolute humidity, UV=Surface solar radiation), OxCGRT Government Response Index, day of the year (day_year), and reproduction number (R_e).

The confounding effect of the OxCGRT index on the association between temperature and R_e is due to the fact that day of the year is associated both with the mean temperature and the OxCGRT index. This implies that adjusting by the OxCGRT index will wholly, or at least partly, remove the confounding effect due to ‘day of the year’. For reassurance that day of the year did not confound the association between meteorological variables and R_e , we additionally controlled for it. The table below shows the p values estimated with multivariable meta-regression multilevel models, which are also adjusted by day of the year. These are consistent with estimates presented in Table 2 of the manuscript. The Figure below shows the similarity of the effect estimated from a model, adjusted also by day of the year and from the model presented in the manuscript, shown in Figure 3.

	From Table 2	Also adjusted by day of the year
Variables	P value*	P value**
Mean temperature (°C)	0.014	0.015
Absolute humidity (g/m ³)	0.036	0.036
Relative humidity (%)	0.058	0.060
Surface solar radiation downwards (J/m ²)	0.208	0.210

Wind speed (m/s)	0.152	0.151
Total precipitation (m)	0.175	0.174

*p values were obtained from multivariable meta-regression multilevel models adjusted by population (log scale), population density (log scale), GDP (log scale), % population > 65 years, PM_{2.5} (log scale), OxCGRT oxford government response index, with cities nested within countries.

**p values were obtained from multivariable meta-regression multilevel models adjusted by population (log scale), population density (log scale), GDP (log scale), % population > 65 years, PM_{2.5} (log scale), OxCGRT oxford government response index and day of the year, with cities nested within countries.

We have included this sensitivity analysis in the supplementary materials, in new Fig S4 and as an additional row in newly labelled Table S5.

R2.2. Given that for most cities, we are now more than a year since the first wave, consideration of the transmission and disease patterns over a longer period of time and/or over several time periods appears to have been possible. While it is true that the less than a full year of data limit analysis of inter-annual patterns, consideration of the relationship between when outbreak window considered and level of preparedness and response may be called for.

Following on from our response to R1.2, for our chosen cross-sectional/ecological approach we saw clear advantages in focusing this study on the early period of the epidemic. This choice allowed us to minimise possible biases coming from factors impacting on R_e (in particular NPIs) developing as the pandemic progressed. These included change of ascertainment methods and strategies, the implementation of strong government interventions (e.g., travel bans, school closures and lockdowns) and ultimately vaccination campaigns. While it is possible that people may have been more prepared in cities with later time windows, which could have driven down transmission as temperatures became warmer in northern hemisphere cities, a sensitivity analysis found that day-of-year was not an important confounder (see response to R2.1). We agree that it would be of interest to look at impact of weather over time. However, this introduces a wide range of

potential sources of bias, for which a cross-sectional study, such as the one we have adopted, is not susceptible to. A different modelling approach would be required to consider time-varying factors impacting on R_e , which is beyond the scope of this study.

R2.3. Another concern, acknowledged by the authors is that most cities are in the northern hemisphere, with 333 cities from 7 countries and 179 from the U.S. a separate analysis for the few southern countries, and for countries that cover a large range of latitudes may be of interest. Italy, in particular with very different initial outbreak patterns may be especially informative, in addition to the U.S.

Our study is based on an ecological analysis and the power of the study is mainly driven by the number of cities and the variability of the exposures across them. Stratified analysis of this nature in a single country (e.g., Italy) could be problematic as there could be low power to detect the association. We performed a sensitivity analysis in the few cities (n=28) in the southern hemisphere and we did not observed associations in this subgroup, but this is probably due to low power. We have now performed an additional sensitivity analysis considering locations with latitude lower than 45 degrees (i.e. excluding cities located in cooler, northern European countries) and the results for temperature are similar to those reported in the main analysis (see table below).

Table: 308 locations with latitude lower than 45 degrees

Variables	P value
Mean temperature (°C)	0.021
Absolute humidity (g/m ³)	0.055
Relative humidity (%)	0.066
Surface solar radiation downwards (J/m ²)	0.211
Wind speed (m/s)	0.028
Total precipitation (m)	0.221

We have included this as an additional row in the supplementary materials in newly labelled Table S5.

We also performed an analysis in 179 US locations and the result for temperature is similar to the main analysis. While this analysis reassures that the US does not indicate a very different pattern to that overall, we acknowledge that it is a cursory analysis and for this reason have not included it in our main results or supplementary materials.

179 locations in US

Variables	P value
Mean temperature (°C)	0.008
Absolute humidity (g/m ³)	0.264
Relative humidity (%)	0.192
Surface solar radiation downwards (J/m ²)	0.955
Wind speed (m/s)	0.765
Total precipitation (m)	0.239

R2.4. With regard to mode of transmission, a consensus appear to have emerged that contact transmission is not an important mechanism, and the impact of meteorological conditions on contact transmission is probably of limited importance for Covid-19 transmission.

Thank you for this comment. Considering this emerging evidence regarding contact transmission, we have removed the following from the discussion:

“In contrast, high relative humidity seems to favour contact transmission of respiratory viruses given that droplets settle more readily on surfaces”

Overall, the paper provides a valuable addition to our understanding of the (limited) impact of weather on Covid-19. The description of the data and the methods used is clear and justified, and the tables and figures are effective. The authors provide good discussion of many of the limitations of their analysis, and the paper is timely, of value, and of interest to readers of this journal.

We thank the reviewer for the positive comment.

Reviewer #3 (Remarks to the Author):

This is the most detailed analysis of the relation between climate and the early spread of COVID-19 to date. The authors have curated a huge dataset of demographic, epidemiological and climatic variables for over 500 local authorities in 26 countries and on 5 continents. Crucially, the analysis account for local variations in reporting and non-pharmaceutical interventions. Given all the issues and limitations of COVID reporting during the first wave of the pandemic, I think the authors have done a very good job, and it seems very unlikely that they could have substantially underestimated the impact of climatic factors.

We thank the reviewer for the positive evaluation of our manuscript.

I don't have any major issues, so this review is more about dotting the i's and crossing the t's.

R3.1. There are two obvious gaps in the sampling of the variables. First, large parts of the world are missing, including Africa, Eastern Europe, the Middle East and South Asia. This presumably stems from missing or unreliable case reports in those regions for the first 4 months of 2020. Looking at Figure 2, the available locations appear to cover a very wide range of climatic values, so it's unlikely that large effects of climate on COVID spread could have been missed. The fact that temperate regions (especially Western Europe and the USA) have been much more intensively sampled might hide some trends in the tails of the temperature and humidity distributions. However, I assume that the inclusion of countries as random factors in the statistical model should help rebalance the data. If anything, the over-sampling in temperate regions might have inflated the association between intermediate temperatures and higher R_e . Second, the authors only considered the initial spread of COVID in each of the locations covered, from January to April 2020. Obviously, there are lots of good reasons to restrict the analysis to the initial period, before sociopolitical factors became too tangled up. However, this may have excluded some countries that reported their first waves later: have you checked? In theory, it would have been interesting to analyse seasonal variations within countries over several months, but I'm pretty sure the data would be too noisy and the models too complex.

We thank the reviewer for the comments. We used the Multi-Country Multi-City (MCC) Collaborative Research Network to retrieve the data for the analysis of this paper. The MCC Network allowed the collection of reliable data at a fine spatial resolution, increasing the accuracy of the variables considered in the analysis. The COVID-19 time-series collected in the 502 locations considered in this analysis until 31 May 2020 represented 44.8% of all cumulative reported COVID-19 cases registered in the John Hopkins University Coronavirus Resource Center by this date (31 May 2020).

As pointed out by the reviewers, we may have excluded countries and locations that experienced the first wave of the outbreak later, but our aim was to collect reliable data for relatively small geographical areas (i.e., cities) and this was feasible using the established MCC Network. Using small-scale geographical units allowed us to reduce measurement error for the exposure (e.g., mean temperature) and the outcome (COVID-19 R_e) and to consider possible socio-demographic covariates (confounders), already collected, summarised, and linked to each city within the network.

Although we reached a good coverage, the sample is not representative of the global spread of COVID-19 with oversampling of temperate regions and countries in Western Europe and USA. Part of the oversampling is, as the reviewer mentioned, considered in the random effect models where the weights are implicitly adjusted to give more weight to less represented countries. We have acknowledged this limitation in the penultimate paragraph of the Discussion section. We also hope that the sensitivity analyses (see newly labelled Table S5), in which we analysed various geographic subsets of the overall data independently, indicated the extent to which we believe our results are generalizable geographically.

We agree with the reviewer that it would be interesting to analyse seasonal variations within countries over a longer period. Within the MCC network we are currently extending the data collection until the end of 2020, for a subsequent time series analysis. In this paper, we focus on the early phase of the pandemic. As noted in our response to R1.2, we saw clear advantages in focusing this study on the early period of the epidemic. This choice allowed us to minimise possible biases coming from factors impacting on R_e (in particular NPIs), which developed as the pandemic progressed. These included change of ascertainment methods and strategies, the appearance of new variants, the implementation of strong government interventions (e.g., lockdown, school closing and travel bans) and ultimately the vaccination campaign. Further work is needed to evaluate if it would be possible to consider all these time-varying confounders in time-series models over a longer period.

Other questions and comments:

R3.2. Although the title and main text only refers to locations as "cities", the locales appear to be a mix of local authorities of various sizes, (e.g. whole départements in France) as briefly acknowledged in the MMethods (l.285). Although I can't think of any obvious issue with the analysis, the use of the more restrictive term "cities" across the manuscript could imply a demographic bias in the selection of data.

In our analysis meteorological variables and socio-demographic covariates are collated and summarised at the city level, while the COVID-19 time-series are defined at the smallest administrative level containing the city. We only included cities for which COVID-19 time series were available for an area in which most of the population resided in that city. We thus consider it reasonable to use the term "city" to refer to our units of analysis.

We have modified the text to acknowledge this detail as follows:

Line 380

"In our analysis, meteorological variables and socio-demographic covariates are collated and summarised at the city level, while the COVID-19 time-series are defined at the smallest administrative level containing the city. We only included cities for which COVID-19 time series were available for an area in which most of the population resided in that city. We therefore refer to our unit of analysis as a city."

R3.3. I don't understand why the authors capped the OxGRT index at all: why would values above 70 introduce substantial "confounding by government interventions"? Why is that a confounding factor if it's included in your model?

We acknowledge that we could have relied on inclusion of the OxCGRT index in regression to control confounding and avoided "capping". However, our concern with that was that the potential for residual confounding would be much stronger if we included the high OxCGRT days. Further, high values of the index are less likely to mean the same across geographies than low values. Inclusion of a linear term for OxCGRT was more likely to be sufficient for a capped OxCGRT than an uncapped one. As described in the supplementary material, in the planning phase of the study we performed some preliminary evaluation of the possible cut-off of the OxCGRT index to identify a compromise to maximize the power of the study (maximising the number of cities included in the analysis) while minimising the risk of confounding bias (decreasing the OxCGRT index cut-off). These preliminary evaluations gave 70 as optimal cut-off. During the analysis we checked the possible residual confounding role of the OxCGRT index by including it as covariate in our model and after observing its strong effect we retained it for all analyses.

We have added the following to the supplementary materials:

Line 17

"During the analysis we checked the possible residual confounding role of the capped OxCGRT index by including the value at the end of the time window (lagged by 10 days) as covariate in our model. After observing its strong effect, we retained it for all analyses."

R3.4. Have the authors considered modelling the effects of OxGRT and climatic variables with a time lag? The values of Re in the short time windows may reflect variations that occurred 2-3 weeks earlier.

We thank the reviewer for the comments that allow us to clarify some aspects of our analysis. As specified in the method section, in this study, we considered the 10 days lagged value of the OxCGRT Government Response Index and for each city we assigned the index at the last day of the specified window for each city (lagged by 10 days). The results are very similar if we considered the mid-point instead of the last day of the city specific time-window. We performed a sensitivity analysis (see newly labelled Table S5) where we considered unlagged values of the OxCGRT Government Response Index and the results are consistent with the main analysis.

We also performed a sensitivity analysis considering 10 days lagged values of the meteorological variables (see newly labelled Supplementary Table S5). The results for temperature are consistent with the main analysis with a similar non-linear association with Re.

R3.5. The estimation of R_e was based on consensus distributions for the generation time and incubation periods. Is there any indication that these distributions may differ among countries, e.g. because of demographic differences?

We tried to retrieve country specific information on generation time and incubation period, but only few countries were able to provide reliable data. We decided to use consensus generation time and incubation periods found in the literature (e.g., Abbott et al., 2020, Lauer et al., 2020) Unlike many studies of this type, we included uncertainty on the summary parameters of our generation and incubation period so that our uncertainty in these distributions was reflected in both the R_e estimates and the final results. However, as the reviewer states, there may be additional between country variation, but this is still an area of research (see link) about which there is little in the literature (see <https://www.medrxiv.org/content/10.1101/2021.05.27.21257936v1>).

REVIEWERS' COMMENTS

Reviewer #1 (Remarks to the Author):

The authors have addressed my points and it's much clearer what they're doing. Now that I do fully understand the work I think they need to add mention of this being an ecological analysis to the abstract, and I think the title is misleading. I read it as a time series type analysis, not a cross sectional study across cities. Also, it's not clear what "COVID-19 seasonality" means. Maybe

"Associations between SARS-CoV-2 transmission and climatological factors through a cross-sectional analysis of 409 cities across 26 countries"

Or

"Associations between SARS-CoV-2 transmission and climatological factors: a cross-sectional analysis of 409 cities across 26 countries"

Reviewer #2 (Remarks to the Author):

My concerns have been addressed thoughtfully in the revisions, and I am happy to recommend this article for publication

Reviewer #3 (Remarks to the Author):

I am satisfied with the authors' response.